# Construction of the national fitness public service satisfaction model in China based on American Customer Satisfaction Index

**Fengqin Tian[1], Jieyou Zhou[1], Fei Liu[2]\***

1 National Fitness Research Center, Guangzhou Sport University, Guangzhou, Guangdong, China,
2 Football School, Guangzhou Sport University, Guangzhou, Guangdong, China

\* 15601632350@163.com

**Data Availability Statement:** The datasets generated for this study are available on request to the corresponding author.

**Funding:** This study is funded by the Guangdong Planning Office of Philosophy and Social Science

## Abstract

### Objective

The national fitness initiative in China is a strategic priority, public satisfaction is a key metric for evaluating the effectiveness of public services. The American Customer Satisfaction Index (ACSI) model has proven a robust tool for evaluating satisfaction with public services. The objective of study was to construct a satisfaction model for national fitness public services in China based on the ACSI framework and to explore the complex relationships among its components.

### Methods

Evaluation dimensions and an item pool were carefully developed by referencing the ACSI model and relevant academic literature. After a panel of experts assessed the initial items pool, the study formed the questionnaire to distribute the residents in Guangzhou, a total of 1,133 valid responses were collected. Data analysis was conducted using SPSS 22.0 and AMOS 16.0 software to evaluate the reliability and validity of the measurement model, the goodness-of-fit of the structural model, and analyze the statistical significance of each path coefficient.

### Results

Public expectation does not directly influence satisfaction (path coefficient 0.039, $p = 0.103$), it exerts a significant and direct influence on perceived quality (path coefficient 0.445, $p = 0.003$), which in turn demonstrably shapes public satisfaction (path coefficient 0.403, $p = 0.005$). Perceived quality also directly influences perceived value (path coefficient 0.735, $p = 0.006$), which in turn significantly impacts public satisfaction (path coefficient 0.554, $p = 0.003$). Public satisfaction directly and significantly influences both public complaints (path coefficient 0.395, $p = 0.003$) and public trust (path coefficient 0.699, $p = 0.003$).

### Conclusion

Perceived quality is the most critical factor influencing public satisfaction, which subsequently affects public complaints and trust. Fitness service providers should consistently

(http://www.gdpplgopss.org.cn). The program name is "Integrated Development of School Football in the Guangdong-Hongkong-Macao Bay Area" (GD24YTY06). We state that any authors didn't receive a salary from the funder, and the funder had no role in study design, data collection and analysis, decision to publish, or preparation of the manuscript.

**Competing interests:** The authors have declared that no competing interests exist.

improve service quality, aligning actual service delivery with public expectations, and enhancing the perceived value for the public. These efforts will bolster the public's satisfaction level.

## Introduction

Currently, China's remarkable economic progress and rising living standards have ignited a widespread desire for physical well-being among its citizens. This burgeoning demand requires the government to provide comprehensive and high-quality public sports service. However, the national fitness public service in China falls short, characterized by inadequate scale and quality, hindering its ability to meet the public's fitness needs [1]. Recognizing this gap, China has embarked on prioritizing the structural reform of national fitness public service supply. A landmark directive, issued in 2022 by the General Office of the Communist Party of China Central Committee and the General Office of the State Council, emphasizes the importance of establishing a national fitness public service system that operates at a higher level of quality and comprehensiveness, and allocate national fitness public service resources based on the public's intrinsic needs [2]. The public, as active participants and beneficiaries of national fitness public service, holds the right to assess all aspects of public sports service. This encompasses the planning, design, and execution of service projects, as well as the manner and quality of service delivery. Public satisfaction with these services is a crucial barometer for gauging the government's service provision efficacy, which offers a potent mechanism for stimulating the reform of the supply side of the public service.

The origins of customer satisfaction theory can be traced back to the 1930s with pioneers in social and experimental psychology, Hoppe (1930) and Lewin (1936), laying the foundational groundwork [3, 4]. Cardozo (1965) introduced the concept of customer satisfaction into the marketing domain [5]. On the whole, satisfaction is a subjective experience, a reflection of the comparison between perceived product functionality, attributes, and pre-existing expectations [6]. When a product or service aligns with (or surpasses) the pre-determined expectations [7], the customer will arise a state of fulfillment and make a positive evaluation of the product or service [8]. Since the 1980s, some sophisticated models that dissect the intricate components of customer satisfaction has been constructed. These models serve as the foundation for nationwide customer satisfaction index evaluations, a critical tool for enhancing the competitiveness of domestic enterprises. For example, Sweden spearheaded this movement with the establishment of the world's inaugural national customer satisfaction barometer in 1989 [9]. the United States launched the American Customer Satisfaction Index (ACSI) in 1994 [10], and the European Union embarked on its first European Customer Satisfaction Index (ECSI) measurement in 1999 [11]. China Institute of Standardization and Tsinghua University collaboratively developed the China customer satisfaction index (CCSI) in 2002, providing a framework for assessing and improving customer experiences within the Chinese market [12].

As customer satisfaction research has generated a wealth of valuable insights, scholars began to explore its applicability within the realm of public service. While the public does not engage in traditional market transactions for public services, their significant contributions through tax payments throughout their lives establish a compelling case for viewing them as "customers" who "consume" the public services [13]. Consequently, the quality and efficacy of public service can be effectively assessed through public satisfaction, which has witnessed a wealth of research. For instance, Kwon et al. (2018) leveraged the public service satisfaction

index to assess user perceptions of Korean libraries [14]. Drawing upon the theoretical foundation of the American Customer Satisfaction Index (ACSI) model, Van Ryzin et al. (2004) conducted a comprehensive investigation of New York City residents' satisfaction with local government services [15]. Li (2021) and Freed (2009) have respectively developed models for evaluating public satisfaction with e-government services in China and the United States [16, 17]. Ochoa-Rico et al. (2024) conducted a comparative analysis of rural and urban residents' satisfaction with public services in Guayaquil [18].

The academic community in China has exhibited significant interest in the provisioning reform of national fitness public service. However, much of the existing research remains focused on the theoretical analysis of national fitness policies and comparative studies of domestic and international fitness systems. Empirical research examining the national fitness public service from the perspective of public satisfaction are still limited. Moreover, there is a deficiency in the availability of a scientific model to accurately assess satisfaction with national fitness services. The ACSI model, recognized for its maturity, has been widely adopted by public sectors worldwide as a reliable tool for assessing public satisfaction. In China, an increasing number of public entities have introduced this model. Therefore, the objective of this study was to develop the dimensions and corresponding indexes for a national fitness service model tailored to China based on the ACSI framework, and further to validate the reliability and validity of the measurement model and the structural model's fit level using survey data. Ultimately, the study elucidated the complex interrelationships among the various dimensions of the model.

## Theory framework

Originating in 1994, the ACSI model is designed, conducted, and analyzed by the National Quality Research Center, Stephen M. Ross School of Business at the University of Michigan. The model's efficacy was demonstrably established through it annually collects data from some 400,000 consumers residing across the United States for more than 400 companies within about 50 consumer industries [19]. In 1999, the US federal government selected the ACSI to be a standard metric for measuring citizen satisfaction. Now, the ACSI measures citizen satisfaction with over 100 services, programs, and websites of federal government agencies [20]. The ACSI model comprises six variables: customer expectation, perceived quality, perceived value, customer satisfaction, customer complaint, and customer loyalty. This model's structure is predicated on causality, the initial three variable–customer expectation, perceived quality, and perceived value–serve as antecedents, shaping customer satisfaction. Customer satisfaction, in turn, impacts customer complaint and loyalty, marking the end of the causal chain.

Given the ACSI model has widespread adopted in governmental public service evaluation. The study constructs a comprehensive framework for evaluating the public's satisfaction with national fitness services based on ACSI model (see Fig 1).

## Public expectation

Public expectation is a psychological phenomenon reflecting an individual's anticipation of achieving a desired outcome or fulfilling a particular need within a specified timeframe based on past experiences. The national fitness public service expectations can be analyzed from three important perspectives: the expectation of comprehensive national fitness public service provision, the expectation of reliable national fitness public service provision, and the expectation of personalized national fitness public service.

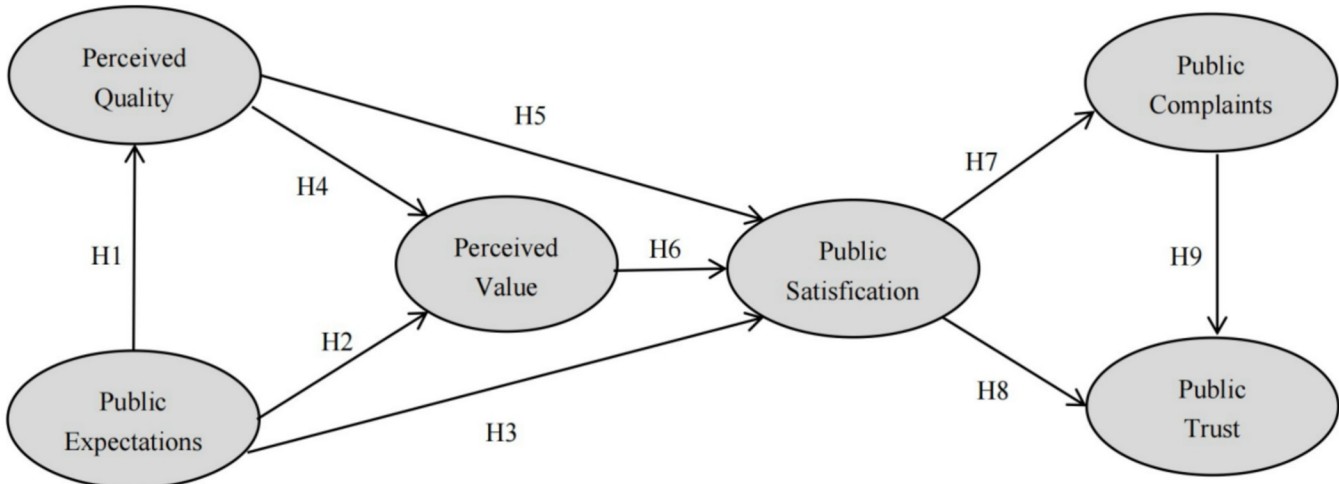

**Fig 1. National fitness public services satisfaction model.**

### Perceived quality

The perceived quality of national fitness public service hinges on the public's subjective experience and evaluation of these services. Based on Gronroos's study (1984) [21], the perceived quality of national fitness public services can be assessed through functional quality. Functional quality refers to the perceived level of service experienced during the interaction itself. This includes, but is not limited to, the quality and accessibility of fitness facilities, the diversity and efficacy of fitness activities, the expertise of fitness instructors, the comprehensiveness of physical testing services, the clarity and accessibility of fitness information, the effectiveness of policy implementation, and the accessibility of complaint channels.

### Perceived value

The perceived value of national fitness public service is primarily assessed through the quality relative to price and vice versa. This value is a subjective evaluation held by the public, reflecting their sentiment about the quality of the service provided and the fairness of its associated costs. It hinges upon the perception of the service's price in comparison to its quality, or conversely, the quality perceived in relation to the price paid.

### Public satisfaction

Public satisfaction arises from a delicate interplay between public expectations and the perceived efficacy of service delivery. According to Fornell's study (1996) [10], public satisfaction can be assessed in three important aspects: overall satisfaction, a comprehensive assessment of the public service as a whole; discrepancy between expectation and reality, the degree to which the actual service delivery falls short of or surpasses public expectations; gap with the ideal, the distance between the existing service and an idealized, perfect offering.

### Public complaints

Public complaints with national fitness services arises from perceived inadequacies in product or service delivery, this can evoke a variety of negative emotional and behavioral responses. These expressions of discontent take three primary forms: informal complaints shared among

personal networks, public commentary disseminated through social media platforms, and formal grievances submitted to relevant authorities.

## Public trust

Public trust with national fitness service emerges from a positive perception of the services and products offered. This trust manifests as confidence in the efficiency and quality of services provided by relevant departments and sports organizations. It is reflected in the willingness to engage in future fitness activities, recommend these services to others, and actively seek out information and knowledge related to fitness.

In the hypothesis model, there are three antecedent variables that significantly impact public satisfaction. Public expectation plays a pivotal role in shaping perceptions of national fitness public service, anticipations regarding future service quality and performance exert a profound influence on how individuals evaluate their experiences. The perceived quality of a national fitness public service upon actual experience directly impacts satisfaction levels. Alignment or exceeding of pre-existing expectations by the actual service delivery results in heightened satisfaction. Furthermore, when individuals perceive the quality of a national fitness public service to be high, coupled with reasonable charges, they experience a sense of perceived gain, leading to enhanced satisfaction. Conversely, a low-quality service, coupled with excessive charges, diminishes perceived value and contributes to dissatisfaction. So, the following hypotheses are proposed:

H1: There is a positive correlation between public expectation and perceived quality.

H2: There is a positive correlation between public expectation and perceived value.

H3: There is a positive correlation between public expectation and public satisfaction.

H4: There is a positive correlation between perceived quality and perceived value.

H5: There is a positive correlation between perceived quality and public satisfaction.

H6: There is a positive correlation between perceived value and public satisfaction.

The satisfaction of national fitness public services can impact public trust and complaints. A satisfied citizenry translates to support for the government departments or sports organizations responsible for delivering these services, while simultaneously diminishing the number of grievances. So, the following hypotheses are proposed:

H7: There is a negative correlation between public satisfaction and public complaints.

H8: There is a positive correlation between public satisfaction and public trust.

H9: There is a negative correlation between public complaints and public trust.

## Methods

Drawing upon the established public satisfaction model and existing research on the supply reform of fitness public service, the study has developed an initial set of items for assessing national fitness service. Table 1 provides the literature sources of constructing these items. To ensure clarity and rationality in the evaluation items, five experts in the field of national fitness were consulted. Their feedback led to the inclusion of two additional items: the degree of openness of fitness venues (facilities) and the safety of fitness venues (facilities). Moreover, these experts emphasized the important role of Social Sports Instructors in the national fitness service, so the study included five items related to these professionals. Subsequently, the study

**Table 1. Sources of measurement items of the national fitness service satisfaction model.**

| Potential variable | Observation item | Item source |
|---|---|---|
| **Public expectations** | PE1: My general expectation of national fitness public service before fitness | Fornell et al. (2006) [22]; |
| | | Zhang (2011) [23]; |
| | | Liu and Zhang (2014) [24]; |
| | PE2: My expectation of the reliability of the national fitness public service before fitness | Liang et al. (2012; 2015) [25, 26]; |
| | PE3: My expectation for national fitness public services to meet individual needs before fitness | Zou (2008) [27]; |
| **Perceive quality** | PQ1: The opening-up degree of fitness venues (facilities) | Fornell et al. (2006) [22]; |
| | | Zhang (2011) [23]; |
| | PQ2: The safety of fitness venues (facilities) | Liu and Zhang (2014) [24]; |
| | PQ3: Adequate number of venues (facilities) | Guo (2020) [28]; |
| | PQ4: The diversity of fitness activities | Wu et al. (2007) [29]; |
| | PQ5: The compatibility of fitness activities with the individual's needs | Li and Xu (2017) [30]; |
| | PQ6: The development level of community fitness activities | Zheng and Zhang (2016) [31]; |
| | | Shen et al. (2018) [32]; |
| | PQ7: The development level of scientific fitness guidance service | Lv and Zhang (2018) [33]; |
| | | Wang et al. (2013) [34]; |
| | PQ8: The development level of community physical health testing service | Zhang et al. (2013) [35]; |
| | | Wang (2016) [36]; |
| | PQ9: The promote level of fitness information (timely, accurate and comprehensive) | Wang et al. (2018) [37]; |
| | | Hao et al. (2018) [38]; |
| | PQ10: The ability of the Government shaping fitness culture | Shi and Dai (2021) [39]; |
| | PQ11: The ability of the Government creating fitness atmosphere | Zhong (2018) [40] |
| | PQ12: The investment level of the Government funds to national fitness | |
| | PQ13: The investment level of the Government using or encouraging social capital to national fitness | |
| | PQ14: The implementation effect of national fitness policy | |
| | PQ15: The participation degree of the citizens in the formulation of fitness policy | |
| | PQ16: The extent to which the citizens participate in the implementation of fitness policies | |
| | PQ17: The convenience of the citizens expressing fitness needs | |
| | PQ18: The handling speed of the citizens' fitness suggestions | |
| | PQ19: The completeness of oversees system of the citizen to the governmental department | |
| | PQ20: The accessibility of complaint channels of the citizen to the governmental department | |
| | PQ21: Adequate number of sports social instructors | |
| | PQ22: The professionalism of sports social instructors | |
| | PQ23: The efficiency of sports social instructors | |
| | PQ24: The pleasure of communicating with sports social instructors | |
| | PQ25: The civilization of sports social instructors | |

(*Continued*)

**Table 1.** (Continued)

| Potential variable | Observation item | Item source |
|---|---|---|
| Perceive value | PV1: The provided quality level of the public fitness service in the free and low pricing time | Fornell et al. (2006) [22]; |
| | | Zhang (2011) [23]; |
| | PV2: The rationality of pricing compared with the quality level of public fitness service | Wu et al. (2007) [29] |
| Public Satisfaction | PS1: I am generally satisfied with the public fitness service I have experienced | Fornell et al. (2006) [22]; |
| | | Guo (2020) [28]; |
| | PS2: The quality of public fitness service meets my expectations | Wu et al. (2007) [29]; |
| | PS3: The quality of public fitness service meets my ideal service | Liu and Chen (2006) [41] |
| Public complaints | PC1: I have the experience of complaining to people around me when I am not satisfied with the public fitness service | Fornell et al. (2006) [22]; |
| | | Liu and Zhang (2014) [24]; |
| | PC2: I have the experience of posting comments to social media when I am not satisfied with the public fitness service | Wu et al. (2007) [29] |
| | PC3: I have the experience of complaining to the relevant departments when I am not satisfied with the public fitness service | |
| Public trust | PT1: The fitness service provided by the government will be better | Fornell et al. (2006) [22]; |
| | | Liu and Zhang (2014) [24]; |
| | PT2: The government will act in the best interest of the citizens' fitness | Guo (2020) [28]; |
| | PT3: I will continue to pay attention to the development dynamics of national fitness | Wu et al. (2007) [29] |
| | PT4: I will take full advantage of the fitness service around me | |
| | PT5: I will recommend others to use fitness service | |

established the "National Fitness Public Service Satisfaction Questionnaire". Public expectations and perceived quality were assessed on a Likert seven-point scale ranging from "very low" to "very high". Perceived value, public satisfaction, public complaints, and public trust were similarly measured using a seven-point scale ranging from "very disapprove" to "very approve".

This study was approved by the Ethics Committee of Guangzhou Sport University, with the ethical review number 2020LCLL-011. A comprehensive survey conducted between December 1st and 30th, 2020, to gather insights from Guangzhou citizens. The survey period coincided with the outbreak of COVID-19, necessitating a hybrid approach to data collection. Using a convenience sampling method, the study delivered online questionnaires and paper questionnaires. The digital questionnaire was distributed via the Questionnaire Star platform (https://www.wjx.cn), and shared in fitness-related WeChat groups. In addition, effective epidemic control of the epidemic in China at the time allowed residents in unaffected areas to resume outdoor activities, following regular nucleic acid tests. This enabled the research team to visit various district national fitness centers across Guangzhou to distribute paper questionnaires. The team successfully coordinated with center managers and staff to rally local residents who were engaged in fitness activities. With residents' consent, paper questionnaires were distributed and collected on-site. The study distributed 100 questionnaires in each of Huadu, Baiyun, and Panyu districts; 200 in each of Liwan and Haizhu districts; 240 in Tianhe district; and 80 in Yuexiu district.

**Table 2. Demographic characteristics of the survey samples (N = 1133).**

| Variable | | Frequency | Percentage (%) |
|---|---|---|---|
| **Gender** | Male | 430 | 38 |
| | Female | 703 | 62 |
| **Age group** | Under 20 years old | 233 | 20.6 |
| | 21–30 years old | 37 | 3.3 |
| | 31–40 years old | 565 | 49.9 |
| | 41–50 years old | 276 | 24.4 |
| | 51–60 years old | 19 | 1.7 |
| | Over 60 years old | 3 | 0.1 |
| **Educational background** | Junior high school and below | 346 | 30.5 |
| | Senior high school | 190 | 16.8 |
| | Secondary vocational and technical schools | 238 | 21.0 |
| | Junior college | 204 | 18.0 |
| | Undergraduate | 130 | 11.5 |
| | Master | 25 | 2.2 |

The study collected 1494 questionnaires, comprising 663 Digital questionnaires and 831 paper responses. To ensure the integrity of the collected data, questionnaires failing to meet specific criteria were deemed invalid. These criteria included: 1) incomplete answer, 2) exhibiting uniform responses across all items, suggesting a lack of thoughtful engagement, and 3) questionnaires completed in an unreasonably short time frame that less than five minutes. A preliminary survey was conducted with 15 postgraduate students specializing in social and leisure sport, established an average completion time of approximately 5 minutes. For online questionnaires, the system automatically tracked the time completing each form. For paper questionnaires, they were distributed in a centralized manner, with research team members monitored completion times on-site, submitted in under 5 minutes were deemed invalid. Finally, 1133 valid responses were identified, resulting in an effective response rate is 75.8%. Of the valid response, 438 were from digital questionnaire, 695 were from paper questionnaires.

The study used SPSS 22.0 software to analyze the demographic characteristic of the survey participants, as detailed in Table 2. To assess the reliability and validity of the measurement models, SPSS 22.0 and AMOS 16.0 software were employed to calculate key metrics, including Factor Loadings, Average Variance Extracted (AVE), Composite Reliability (CR), and Cronbach's alpha coefficient. Additionally, AMOS 16.0 software was used to evaluate the fit level of the structural model by calculating various fit indices, including the chi-square to degrees of freedom ($\chi^2$/df) ratio, Goodness of Fit Index (GFI), Adjusted Goodness of Fit Index (AGFI), Incremental Fit Index (IFI), Tucker-Lewis Index (TLI), Comparative Fit Index (CFI), Normed Fit Index (NFI), Relative Fit Index (RFI), Parsimonious Normed Fit Index (PNFI), and Parsimonious Goodness of Fit Index (PGFI).

## Results

### Measurement model

The National Fitness public service satisfaction Model is composed of six structural components, each representing a key measurement model: public expectation, perceived quality, perceived value, public satisfaction, public complaint, and public trust. Confirmatory Factor Analysis using AMOS 16.0 was employed to assess the reliability and validity of these

measurement models, there are several critical indicators: 1) Factor loadings represent the strength of the relationship between each observed variable and its corresponding latent construct. A factor loading exceeding 0.71 is considered excellent, above 0.63 is very good [42]. 2) AVE quantifies the extent to which a latent variable explains the variance in its associated indicators. A higher AVE value indicates stronger convergence validity, a threshold of 0.5 for AVE is commonly considered acceptable [43]. 3) A higher CR value reflects greater internal consistency, indicating that the items within the latent variable are coherently measuring the intended construct. Generally, a CR value exceeding 0.7 is considered to be a good reliability [44]. 4) Cronbach's alpha coefficient exceeding 0.80 is generally considered indicative of strong internal consistency [45].

Table 3 reveals that the all items' factor loadings exceed the threshold of 0.63. Moreover, AVE values, CR values, and Cronbach's alpha coefficients for the dimensions of public expectation, perceived quality, perceived value, public satisfaction, public complaint, and public trust all surpass 0.5, 0.7, and 0.8, respectively. These results demonstrate the internal reliability and validity of the measurement model.

## Structural model

AMOS 16.0 software was employed to analysis fit level of the structural model. As Table 4 reveals, the initial model's fit indices fell short of established benchmarks. Following the methodology outlined by Srbom (1989) [46], the modification index (MI) was utilized to guide model refinement. A total number of fifteen modification were made in model (see Fig 2), the modified structural model has demonstrably improved fit index values. Notably, the sample size exerts a significant influence on the $\chi^2$/df, GFI, and AGFI values [47], necessitating incorporates other indicators for a comprehensive evaluation. Table 4 demonstrates that these other indicators are higher than recommended values, signifying the modified model is a good fit model.

The path coefficient is a pivotal metric for quantifying the direct influence between constructs, a more substantial path coefficient signifies a stronger direct relationship between these constructs. The study employs the Bootstrap method with 500 resamples to determine the significance level of the model's coefficients. Notably, the overall model fit and parameter estimations using the Bootstrap method remain consistent between the modified models. The bilateral significance test results for each path are presented in Table 5, the path PE→PQ, PQ→PV, PE→PV, PV→PS, PQ→PS, PS→PC, and PS→PT exhibited statistically significant (p<0.05). Conversely, the path PE→PS and PC→PT did not demonstrate statistical significance. These findings provide strong support for hypotheses H1, H2, H4, H5, H6, H7, and H8, while failing to validate hypotheses H3 and H9.

## Discussion

### The antecedents of public satisfaction

Empirical analysis reveals that public satisfaction has three primary determinants: public expectation, perceived quality, and perceived value. While public expectation exerts no direct, statistically significant influence on public satisfaction (path coefficient of 0.039, p>0.05), it significantly impacts perceived quality (path coefficient of 0.445, p<0.01), which in turn significantly impacts public satisfaction. Public expectation is commonly understood as the anticipated level of service, product, or policy performance, serving as a comparative benchmark. Service quality is essentially a cognitive assessment of the performance of a given service's attributes (Gallarza et al., 2011; Zeithaml et al., 2009) [48, 49]. Public sports services fall under the participatory consumer health/fitness services (Chelladurai, 1992;1994) [50, 51], participants

**Table 3. Test results of the reliability and validity of the measurement model.**

| Construct | Items | Factor loading | AVE | CR | Cronbach's alpha |
|---|---|---|---|---|---|
| **Public expectations** | PE1 | 0.833 | 0.7309 | 0.8906 | 0.889 |
| | PE2 | 0.900 | | | |
| | PE3 | 0.830 | | | |
| **Perceived quality** | PQ1 | 0.679 | 0.7174 | 0.9844 | 0.984 |
| | PQ2 | 0.667 | | | |
| | PQ3 | 0.788 | | | |
| | PQ4 | 0.791 | | | |
| | PQ5 | 0.802 | | | |
| | PQ6 | 0.840 | | | |
| | PQ7 | 0.862 | | | |
| | PQ8 | 0.865 | | | |
| | PQ9 | 0.859 | | | |
| | PQ10 | 0.859 | | | |
| | PQ11 | 0.835 | | | |
| | PQ12 | 0.840 | | | |
| | PQ13 | 0.867 | | | |
| | PQ14 | 0.894 | | | |
| | PQ15 | 0.887 | | | |
| | PQ16 | 0.890 | | | |
| | PQ17 | 0.859 | | | |
| | PQ18 | 0.891 | | | |
| | PQ19 | 0.898 | | | |
| | PQ20 | 0.897 | | | |
| | PQ21 | 0.892 | | | |
| | PQ22 | 0.873 | | | |
| | PQ23 | 0.881 | | | |
| | PQ24 | 0.876 | | | |
| | PQ25 | 0.831 | | | |
| **Public satisfaction** | PS1 | 0.931 | 0.7239 | 0.8872 | 0.953 |
| | PS2 | 0.943 | | | |
| | PS3 | 0.929 | | | |
| **Perceived value** | PV1 | 0.882 | 0.7868 | 0.8807 | 0.881 |
| | PV2 | 0.929 | | | |
| **Public complaint** | PC1 | 0.801 | 0.7744 | 0.9111 | 0.915 |
| | PC2 | 0.944 | | | |
| | PC3 | 0.912 | | | |
| **Public trust** | PT1 | 0.839 | 0.6976 | 0.9016 | 0.941 |
| | PT2 | 0.866 | | | |
| | PT3 | 0.884 | | | |
| | PT4 | 0.907 | | | |
| | PT5 | 0.873 | | | |

engaged in physical activity act as co-producers of the service experience. They rely on service providers to furnish appropriate facilities and equipment, and in certain cases, professional guidance by service providers [52]. In service quality assessment, satisfaction is affected on the expectation-perception gap, namely the discrepancy between the actual service quality experienced and the expectations previously set. Satisfaction is achieved when the perceived

Table 4. Test results of the original model and the modified model.

| Fit indices | Recommended value | Initial model | Modified model |
|---|---|---|---|
| $x^2$/df | ≤3.00 | 12.033 | 7.146 |
| RMSEA | 0.05 to 0.08 | 0.099 | 0.074 |
| GFI | ≥0.90 | 0.616 | 0.775 |
| AGFI | ≥0.80 | 0.570 | 0.744 |
| NFI | ≥ 0.90 | 0.843 | 0.909 |
| RFI | ≥0.90 | 0.833 | 0.901 |
| IFI | ≥0.90 | 0.854 | 0.920 |
| TLI | ≥0.90 | 0.845 | 0.914 |
| CFI | ≥0.90 | 0.854 | 0.920 |
| PNFI | ≥0.5 | 0.792 | 0.838 |
| PGFI | ≥0.5 | 0.551 | 0.681 |

performance meets or exceeds these expectations [53]. This highlights the critical role of perceived quality in shaping public satisfaction.

The study reveals that though a statistically significant correlation (P = 0.036) between public expectation and perceived value, but the influence is relatively minor, as evidenced by the small path coefficient of 0.067. In contrast, perceived quality exhibits a significant impact on perceived value (path coefficient 0.735, p<0.01), which in turn significantly affects public satisfaction (path coefficient 0.554, P<0.01). This finding highlights the critical role of perceived value in fostering public satisfaction. Perceived value is essentially how customers perceive a product or service, in which customers evaluate costs versus benefits [54]. High-quality services or products—characterized by attributes such as reliability, effectiveness, and superiority —tend to lower perceived costs relative to the benefits received. Therefore, the higher the quality of a service, the greater the perceived value, and increasing the likelihood of public satisfaction. This positive correlation among quality, perceived value, and satisfaction has been substantiated by numerous studies, such as Yu et al. (2014) [55], Hu et al. (2009) [56], Kuo et al. (2009) [57].

In China, national fitness public services are not universally accessible without payment, with many venues (or facilities) requiring payment for public access. For example, the without charges time of National Fitness Center in Tianhe District, Guangzhou, is from Tuesday to Sunday between 9:00 AM and 12:00 PM, the rest of the time is charged. Baiyun District's fitness center offers free badminton court access on weekdays only from 2:00 PM to 4:00 PM. The football field and basketball field are only available for free use throughout the week from 7:00 AM to 10:00 AM. In the Huadu District Sports Center, its tennis court charges 50 yuan per hour for usage before 5:00 PM and 75 yuan per hour for usage after 5:00 PM. Ultimately, the public's perception of value is directly correlated to the quality of service provided. If the national fitness service quality is higher, the public will have a higher perceived value, that is, value for money.

## The outcomes of public satisfaction

Public trust and public complaints are inextricably linked to public satisfaction. Public satisfaction has a significant direct impact on public complaints (path coefficient of 0.395, P<0.01). Oliver (1980) defines satisfaction as a subjective psychological response arising from experiences and emotional reactions associated with a particular event or product [58]. Customer complaint behavior, as articulated by Singh and Howell (1985) [59], encompasses all behavioral and non-behavioral responses that stem from dissatisfaction with a purchase event and

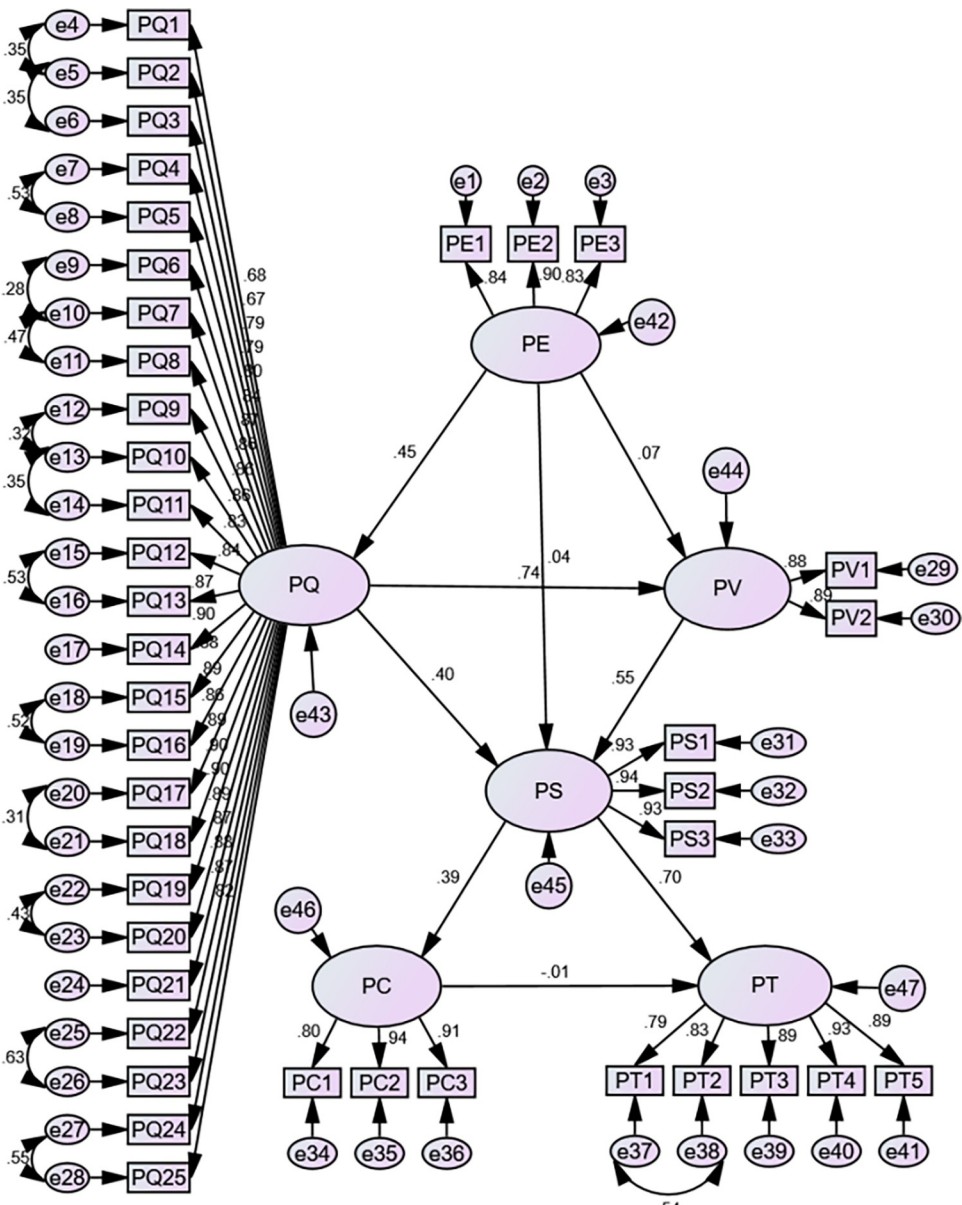

**Fig 2. Modified national fitness public service satisfaction model.**

involve negative communication about the event. Public satisfaction also significantly impacts public trust, with a more pronounced effect, indicated by a path coefficient of 0.699 (P<0.01). This means that when the public is satisfied with public services, they are more likely to trust the institutions or organizations that provide those services. This positive relationship has been confirmed by many studies, such as Welch et al. (2005) [60], Lanin and Hermanto (2019) [61]. The study found that public complaints have no direct effect on public trust (path coefficient = -0.008, P>0.05). This could be because the complaints examined in this study were primarily behavioral outcomes related to specific events or experiences. Whereas trust encompasses a broader evaluation of reliability, integrity, and consistent performance over time. As a result, complaints about specific events may not significantly influence the overall public trust in government services. In summary, high levels of public satisfaction reduce

**Table 5. Significance test of path coefficient.**

| Path | Estimate | P | Hypotheses results |
|---|---|---|---|
| H1: PE→PQ | 0.445 | 0.003 | Supported |
| H2: PE→PV | 0.067 | 0.036 | Supported |
| H3: PE→PS | 0.039 | 0.103 | Rejected |
| H4: PQ→PV | 0.735 | 0.006 | Supported |
| H5: PQ→PS | 0.403 | 0.005 | Supported |
| H6: PV→PS | 0.554 | 0.003 | Supported |
| H7: PS→PC | 0.395 | 0.003 | Supported |
| H8: PS→PT | 0.699 | 0.003 | Supported |
| H9: PC→PT | -0.008 | 0.785 | Rejected |

Note: PE = Public Expectations, PQ = Perceived Quality, PV = Perceived Value, PS = Public Satisfaction, PC = Public Complaints, PT = Public Trust

complaints about public sports services and foster trust in the government's national fitness initiatives.

## Limitations and future research

This study has several limitations. Firstly, using convenience sampling method may bias the sample towards easily accessible populations. For example, online questionnaires might attract individuals who frequently use the internet, while paper questionnaires might draw more individuals who often engage in outdoor activities. The representativeness of the sample and are somewhat constrained. As illustrated in Table 2, there is a low percentage of respondents in the age groups 21–30, 51–60, and over 60 years old. Future studies should control the potential biases introduced by convenience sampling.

Secondly, China's vast geographical expanse and economic disparities lead to variations in the level of public services, which in turn influence residents' perceptions of national fitness services. Owing to the varying in epidemic control measures across different regions in China, this study focused its investigation on Guangzhou, the generalizability of the findings is limited to some extent. To enhance the applicability of the results, future research should consider conducting a nationwide survey to enable broader validation of the model presented here.

Thirdly, to response the challenges posed by the degradation of survey environments and rising execution costs, the use of hybrid surveys—combining multiple data collection methods within a single project—has become common practice in the United States and Europe [62, 63]. Research indicates that the mode of information delivery and the presence of interviewers can influence respondents' answering behaviors [62]. In this study, the paper questionnaire yielded a higher effective response rate compared to the digital version, likely because respondents were more diligent when completing a physical questionnaire under the researcher supervision. In addition, national fitness centers in China are predominantly government-operated, the involvement of fitness center managers may have led participants to perceive the survey as more official. Nonetheless, this study applied a uniform criterion for discarding invalid questionnaires, ensuring consistency and quality across both online and paper-based surveys.

## Conclusion

The national fitness public service satisfaction model in China incorporates six key variables: public expectations, perceived quality, perceived value, public satisfaction, public complaints,

and public trust, along with 41 measurable indicators. Among these variables, public expectation directly affects perceived quality, which in turn directly affects perceived value and public satisfaction. Public satisfaction then directly impacts public complaints and public trust. In the model, perceived quality plays a central role.

The most important practical implication of this study is that government departments should prioritize improving the quality of national fitness public services, such as enhancing the accessibility of fitness venues, upgrading the professional competence of fitness instructors, expanding efforts to offer free or low-cost access to venues, and strengthening channels for complaints and oversight. Enhancing national fitness service quality will enable services to better align with public expectations, boost the perceived value and satisfaction, reduce complaints, and ultimately build greater public trust.

## Supporting information

**S1 Dataset.**
(XLS)

## Acknowledgments

The authors would like to thank all participants who participated in the study's survey.

## Author Contributions

**Conceptualization:** Fei Liu.

**Formal analysis:** Fengqin Tian.

**Funding acquisition:** Fei Liu.

**Investigation:** Fengqin Tian.

**Methodology:** Fei Liu.

**Resources:** Jieyou Zhou.

**Software:** Fei Liu.

**Supervision:** Fei Liu.

**Validation:** Jieyou Zhou.

**Writing – original draft:** Fengqin Tian.

**Writing – review & editing:** Jieyou Zhou.

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
