## [Decision Letter · Decision Letter 0]

1 Dec 2024

PONE-D-24-41589Construction of the national fitness public service satisfaction model in China based on American Customer Satisfaction IndexPLOS ONE

Dear Dr. Liu,

Thank you for submitting your manuscript to PLOS ONE. After careful consideration, we feel that it has merit but does not fully meet PLOS ONE’s publication criteria as it currently stands. Therefore, we invite you to submit a revised version of the manuscript that addresses the points raised during the review process.

We look forward to receiving your revised manuscript.

Kind regards,

Reza Rostamzadeh

Academic Editor

PLOS ONE

Reviewers' comments:

Reviewer's Responses to Questions

**Comments to the Author**

1. Is the manuscript technically sound, and do the data support the conclusions?

Reviewer #1: Yes

Reviewer #2: Partly

Reviewer #3: Yes

Reviewer #4: Yes

2. Has the statistical analysis been performed appropriately and rigorously? 

Reviewer #1: Yes

Reviewer #2: No

Reviewer #3: Yes

Reviewer #4: Yes

3. Have the authors made all data underlying the findings in their manuscript fully available?

Reviewer #1: Yes

Reviewer #2: No

Reviewer #3: Yes

Reviewer #4: Yes

4. Is the manuscript presented in an intelligible fashion and written in standard English?

Reviewer #1: Yes

Reviewer #2: Yes

Reviewer #3: Yes

Reviewer #4: Yes

5. Review Comments to the Author

Reviewer #1: Originality

1. Originality: Does the paper contain new and significant information adequate to justify

publication?

The Paper contain valuable information as considered significant for the justification and publication

The Paper contains valuable information as considered significant for the justification and publication.

Relationship to Literature:

This paper demonstrates a thorough understanding of the subject matter. Citations are also appropriate in a series of works. According to the review, substantial work is part of research.

Methodology:

The author supports the intellectual approach. The use of a survey method combined with an appropriate method result in a well-designed paper.

Results

During the detail review, it was discovered that each indicator of the subject under study is adequately presented. This reflected how each variable in the paper was interconnected. Under control variables analysis can be expanded.

Implications for research, practice and/or society:

The paper can be used to investigate problems and identify areas that are relevant to the subject. The conclusion and findings were based on the data provider’s applicability. Due to the use of long sentences, the sentence structure is unclear. This expression causes confusion for the average reader. However, the case is presented clearly due to the data and scientific approach used. The research technically implies field-specific terms.

Reviewer #2: The study explores an interesting topic. However, there are significant issues should be addressed.

The manuscript primarily applies the ACSI model without offering significant modifications or extensions to suit the context of national fitness services in China. The lack of theoretical innovation limits the paper's contribution to academic literature.

The sampling strategy is inadequately detailed. While 1133 valid responses were collected, the demographic distribution (e.g., age, gender, education) is imbalanced, which could bias the results.

The use of a hybrid data collection method (online and paper-based) introduces potential inconsistencies in response quality, which are not addressed in the discussion.

The study's focus on Guangzhou, China, restricts its applicability to other regions or countries. The findings, while useful for local policymakers, lack broader relevance unless comparisons or implications for other regions are discussed.

While the results demonstrate significant pathways, certain rejected hypotheses (e.g., public expectation directly influencing public satisfaction) are not adequately explained. The discussion could delve deeper into why these relationships failed to materialize and what that implies for theory and practice.

The practical recommendations are too general. For example, suggesting improvements in perceived quality and perceived value does not provide actionable strategies for policymakers or service providers.

Reviewer #3: Dear Authors,

There is a minor issue with the manuscript such as language (line 47: replace "is" with "was").

I congratulate you on the manuscript because the development of national customer satisfaction indices represents an important step towards addressing the gap between what we know and what we need to know. ACSI represents a uniform system for evaluating, comparing, and - ultimately - enhancing customer satisfaction across firms, industries and nations. Other nations such as China, are now adopting the same approach.

Regards.

Reviewer #4: Abstract:

• Consider enhancing the abstract by including specific details for better clarity and impact:

1. Briefly provide the rationale for constructing the national fitness public service satisfaction model to help readers quickly understand its significance and relevance.

2. Incorporate concrete values and statistical significance (e.g., path coefficients, p-values) for key results, enabling readers to grasp the quantitative aspects of the findings at a glance.

3. Ensure the conclusion succinctly summarizes the core findings while offering specific and actionable suggestions based on the results.

Introduction:

• Consider revising the stated objectives and hypotheses to align with the study's main findings, particularly the reliability and validity of the measurement model.

• It is recommended to clearly articulate the rationale for using the ACSI model in this study, further reinforcing the research’s background and motivation.

Methodology:

• The Methods section states that a total of 1,494 questionnaires were collected, comprising both digital (x%) and paper-based (x%) responses. Of these, 1,133 were ultimately included as valid responses. To enhance the transparency and rigor of the methodology, please provide a breakdown of the valid questionnaires by type (e.g., percentage or number of valid digital versus paper-based responses).

• Additionally, one of the exclusion criteria mentioned is a completion time of less than five minutes for digital questionnaires. It would be beneficial to include a rationale for this specific threshold to justify its appropriateness in assessing response quality. Furthermore, please clarify whether similar criteria were applied to paper-based questionnaires, and if so, how completion time was assessed to ensure consistency across both formats.

• Consider revising or expanding the statistical analysis to provide a more comprehensive overview of the techniques employed in this section.

Results:

• For clarity, consider adding the meaning of any abbreviations used in the tables directly within the table captions or footnotes.

Discussion:

• Consider discussing the reliability and validity results more comprehensively. For instance, address any observed lower values in factor loadings or AVE and their potential implications.

• Lines 290–305: The authors are encouraged to strengthen this section by incorporating references to related studies or theories rather than relying primarily on examples. This approach will provide a more robust theoretical framework and better situate the findings within the context of existing literature.

• Please consider acknowledging the limitations of this study that may link to the future study.

Conclusion:

• The conclusion should provide a concise summary of the study’s key findings while offering clear suggestions and specific directions for future research. Avoid repeating details from the Methods and Results sections to maintain a focused and impactful conclusion.

6. PLOS authors have the option to publish the peer review history of their article (what does this mean?). If published, this will include your full peer review and any attached files.

Reviewer #1: No

Reviewer #2: No

Reviewer #3: **Yes: **Ferhat Esatbeyoglu

Reviewer #4: No

---

## [Author Response · Author response to Decision Letter 0]

14 Dec 2024

Dear editor and reviewer:

We extend our sincere gratitude for evaluating our research. According to your esteemed commentary, we have undertaken substantial revisions, highlighted in red within the revised manuscript. We are confident that these revisions have improved the quality and overall contribution of our research. We have crafted a comprehensive point-by-point response to address reviewer’s comments.

The academic editor comments

1.Please ensure that your manuscript meets PLOS ONE’s style requirements, including those for file naming. 

2.Please include your full ethics statement in the ‘Methods’ section of your manuscript file. In your statement, please include the full name of the IRB or ethics committee who approved or waived your study, as well as whether or not you obtained informed written or verbal consent. If consent was waived for your study, please include this information in your statement as well.

Response

1.We have modified the format of the paper to align with the style guidelines of PLOS ONE.

2.We have added an ethical statement to the methods section of the revised manuscript. See lines 229-230 of the revised manuscript. 

Reviewer 1 comments

The paper can be used to investigate problems and identify areas that are relevant to the subject. The conclusion and findings were based on the data provider’s applicability. Due to the use of long sentences, the sentence structure is unclear. This expression causes confusion for the average reader. However, the case is presented clearly due to the data and scientific approach used. The research technically implies field-specific terms.

Response

In the revised manuscript, we have simplified and clarified any lengthy or unclear sentences.

Reviewer 2 comments

The study explores an interesting topic. However, there are significant issues should be addressed.

1. The manuscript primarily applies the ACSI model without offering significant modifications or extensions to suit the context of national fitness services in China. The lack of theoretical innovation limits the paper's contribution to academic literature.

Response

As the reviewer noted, this paper constructs a satisfaction model for China’s national fitness services based on the ACSI framework, which may lack significant theoretical innovation. In China, despite the national fitness being a strategic priority, there is still a notable absence of scientific models to assess satisfaction with these services. The ACSI model has gained increasing recognition and application in evaluating the satisfaction with non-sports public services in China. Therefore, the satisfaction model developed in this study has practical significance. It can guide the reform of the supply side of national fitness, and serve as a foundation for further in-depth research by scholars in China.

2.The sampling strategy is inadequately detailed. While 1133 valid responses were collected, the demographic distribution (e.g., age, gender, education) is imbalanced, which could bias the results.

Response

Firstly, we have included a more detailed sampling strategy in the revised manuscript. See lines 232-242 of the revised manuscript. The specific modifications are outlined as follows:

Using a convenience sampling method, the study delivered online questionnaires and paper questionnaires. The digital questionnaire was distributed via the Questionnaire Star platform (https://www.wjx.cn), and shared in fitness-related WeChat groups. In addition, effective epidemic control of the epidemic in China at the time allowed residents in unaffected areas to resume outdoor activities, following regular nucleic acid tests. This enabled the research team to visit various district national fitness centers across Guangzhou to distribute paper questionnaires. The team successfully coordinated with center managers and staff to rally local residents who were engaged in fitness activities. With residents’ consent, paper questionnaires were distributed and collected on-site. The study distributed 100 questionnaires in each of Huadu, Baiyun, and Panyu districts; 200 in each of Liwan and Haizhu districts; 240 in Tianhe district; and 80 in Yuexiu district. 

Secondly, convenience sampling using in the study may have resulted in an imbalance in the demographic distribution of the sample. This limitation has been acknowledged in the revised manuscript. See lines 374-380 of the revised manuscript. The specific modifications are outlined as follows:

Using convenience sampling method may bias the sample towards easily accessible populations. For example, online questionnaires might attract individuals who frequently use the internet, while paper questionnaires might draw more individuals who often engage in outdoor activities. The representativeness of the sample and are somewhat constrained. As illustrated in Table 2, there is a low percentage of respondents in the age groups 21-30, 51-60, and over 60 years old. Future studies should control the potential biases introduced by convenience sampling. 

3.The use of a hybrid data collection method (online and paper-based) introduces potential inconsistencies in response quality, which are not addressed in the discussion.

Response

In the “Limitations and Future Research” section of the revised manuscript, we have provided an in-depth discussion on the potential inconsistencies in response quality that could arise from employing a hybrid method. See lines 387-397 of the revised manuscript. The specific modifications are outlined as follows:

To response the challenges posed by the degradation of survey environments and rising execution costs, the use of hybrid surveys—combining multiple data collection methods within a single project—has become common practice in the United States and Europe. Research indicates that the mode of information delivery and the presence of interviewers can influence respondents’ answering behaviors. In this study, the paper questionnaire yielded a higher effective response rate compared to the digital version, likely because respondents were more diligent when completing a physical questionnaire under the researcher supervision. In addition, national fitness centers in China are predominantly government-operated, the involvement of fitness center managers may have led participants to perceive the survey as more official. Nonetheless, this study applied a uniform criterion for discarding invalid questionnaires, ensuring consistency and quality across both online and paper-based surveys.

4.The study’s focus on Guangzhou, China, restricts its applicability to other regions or countries. The findings, while useful for local policymakers, lack broader relevance unless comparisons or implications for other regions are discussed.

Response

Owing to the varying in epidemic control measures across different regions in China, this study focused its investigation on Guangzhou, which limits the generalizability of its findings to some extent. This limitation is discussed in the “Limitations and Future Studies” section of the revised manuscript. See lines 381-386 of the revised manuscript. The specific modifications are outlined as follows:

China’s vast geographical expanse and economic disparities lead to variations in the level of public services, which in turn influence residents’ perceptions of national fitness services. Owing to the varying in epidemic control measures across different regions in China, this study focused its investigation on Guangzhou, the generalizability of the findings is limited to some extent. To enhance the applicability of the results, future research should consider conducting a nationwide survey to enable broader validation of the model presented here.

5. While the results demonstrate significant pathways, certain rejected hypotheses (e.g., public expectation directly influencing public satisfaction) are not adequately explained. The discussion could delve deeper into why these relationships failed to materialize and what that implies for theory and practice.

Response

In the discussion section of the revised manuscript, the two rejected assumptions (public expectation → public satisfaction; public complaints → public trust) were discussed in detail. See lines 320-321, 327-331 and 365-372 of the revised manuscript. The specific modifications are outlined as follows:

Public expectation exerts no direct, statistically significant influence on public satisfaction (path coefficient of 0.039, p>0.05), it significantly impacts perceived quality (path coefficient of 0.445, p<0.01), which in turn significantly impacts public satisfaction. Public expectation is commonly understood as the anticipated level of service, product, or policy performance, serving as a comparative benchmark. In service quality assessment, satisfaction is affected on the expectation-perception gap, namely the discrepancy between the actual service quality experienced and the expectations previously set. Satisfaction is achieved when the perceived performance meets or exceeds these expectations (Zhang et al.,2022). This highlights the importance of perceived quality in shaping public satisfaction.

The study found that public complaints have no direct effect on public trust (path coefficient = -0.008, P>0.05). This could be because the complaints examined in this study were primarily behavioral outcomes related to specific events or experiences. Whereas trust encompasses a broader evaluation of reliability, integrity, and consistent performance over time. As a result, complaints about specific events may not significantly influence the overall public trust in government services. In summary, high levels of public satisfaction reduce complaints about public sports services and foster trust in the government’s national fitness initiatives. 

6. The practical recommendations are too general. For example, suggesting improvements in perceived quality and perceived value does not provide actionable strategies for policymakers or service providers.

Response

In the conclusion section of the revised manuscript, we have outlined the practical recommendations in detail. See lines 405-411 of the revised manuscript. The specific modifications are outlined as follows:

The most important practical implication of this study is that government departments should prioritize improving the quality of national fitness public services, such as enhancing the accessibility of fitness venues, upgrading the professional competence of fitness instructors, expanding efforts to offer free or low-cost access to venues, and strengthening channels for complaints and oversight. Enhancing service quality will enable national fitness services to better align with public expectations, boost the perceived value and satisfaction, reduce complaints, and ultimately build greater public trust.

Reviewer 3 comment

There is a minor issue with the manuscript such as language (line 47: replace "is" with "was").

Response

We have fixed this language error. See line 49 of the revised manuscript.

Reviewer 4 comments

Abstract:

• Consider enhancing the abstract by including specific details for better clarity and impact:

1. Briefly provide the rationale for constructing the national fitness public service satisfaction model to help readers quickly understand its significance and relevance.

2. Incorporate concrete values and statistical significance (e.g., path coefficients, p-values) for key results, enabling readers to grasp the quantitative aspects of the findings at a glance.

3. Ensure the conclusion succinctly summarizes the core findings while offering specific and actionable suggestions based on the results.

Response

In the abstract section of the revised manuscript, we have briefly provided the rationale for constructing the national fitness public service satisfaction model, included concrete values and statistical significance in the results, and revised the conclusion to summarize the study’s core findings. Additionally, we have structured the abstract into four distinct sections: objective, methods, results, and conclusion. See lines 45-70 of the revised manuscript. 

Introduction:

1. Consider revising the stated objectives and hypotheses to align with the study's main findings, particularly the reliability and validity of the measurement model.

2. It is recommended to clearly articulate the rationale for using the ACSI model in this study, further reinforcing the research’s background and motivation.

Response

In the introduction of the revised manuscript, we have refined the objectives and hypotheses, and clearly articulated the rationale for using the ACSI model. See lines 126-134 of the manuscript. The details of the modifications are outlined as follows:

Moreover, there is a deficiency in the availability of a scientific model to accurately assess satisfaction with national fitness services. The ACSI model, recognized for its maturity, has been widely adopted by public sectors worldwide as a reliable tool for assessing public satisfaction. In China, an increasing number of public entities have introduced this model. Therefore, the objective of this study was to develop the dimensions and corresponding indexes for a national fitness service model tailored to China based on the ACSI framework, and further to validate the reliability and validity of the measurement model and the structural model’s fit level using survey data. Ultimately, the study elucidated the complex interrelationships among the various dimensions of the model.

Methodology:

1. The Methods section states that a total of 1,494 questionnaires were collected, comprising both digital (x%) and paper-based (x%) responses. Of these, 1,133 were ultimately included as valid responses. To enhance the transparency and rigor of the methodology, please provide a breakdown of the valid questionnaires by type (e.g., percentage or number of valid digital versus paper-based responses).

2. Additionally, one of the exclusion criteria mentioned is a completion time of less than five minutes for digital questionnaires. It would be beneficial to include a rationale for this specific threshold to justify its appropriateness in assessing response quality. Furthermore, please clarify whether similar criteria were applied to paper-based questionnaires, and if so, how completion time was assessed to ensure consistency across both formats.

3. Consider revising or expanding the statistical analysis to provide a more comprehensive overview of the techniques employed in this section.

Response

In the revised manuscript, we have included a breakdown of the valid questionnaires by type, and clarified the criteria for deeming a digital questionnaire invalid if it is completed in less than 5 minutes, a threshold that also applies to paper questionnaires. Furthermore, we have expanded the overview of the statistical analysis techniques employed in this section. See lines 247-264 of the revised manuscript. The specific modifications are outlined as follows:

A preliminary survey was conducted with 15 postgraduate students specializing in social and leisure sport, established an average completion time of approximately 5 minutes. For online questionnaires, the system automatically tracked the time completing each form. For paper questionnaires, they were distributed in a centralized manner, research team members monitored completion times on-site, submitted in under 5 minutes were deemed invalid. Finally, 1133 valid responses were identified, resulting in an effective response rate is 75.8%. Of the valid response, 438 were from digital questionnaire, 695 were from paper questionnaires. 

The study used SPSS 22.0 software to analyze the demographic characteristic of the survey participants, as detailed in Table 2. To assess the reliability and validity of the measurement models, SPSS 22.0 and AMOS 16.0 software were used to calculate key metrics, including Factor Loadings, Average Variance Extracted (AVE), Composite Reliability (CR), and Cronbach’s alpha coefficient. Additionally, AMOS 16.0 software was used to evaluate the fit level of the structural model by calculating various fit indices, including the chi-

---

## [Decision Letter · Decision Letter 1]

2 Jan 2025

Construction of the national fitness public service satisfaction model in China based on American Customer Satisfaction Index

PONE-D-24-41589R1

Dear Dr. Liu,

We’re pleased to inform you that your manuscript has been judged scientifically suitable for publication and will be formally accepted for publication once it meets all outstanding technical requirements.

Kind regards,

Reza Rostamzadeh

Academic Editor

PLOS ONE

Additional Editor Comments (optional):

Reviewers' comments:

Reviewer's Responses to Questions

**Comments to the Author**

1. If the authors have adequately addressed your comments raised in a previous round of review and you feel that this manuscript is now acceptable for publication, you may indicate that here to bypass the “Comments to the Author” section, enter your conflict of interest statement in the “Confidential to Editor” section, and submit your "Accept" recommendation.

Reviewer #3: All comments have been addressed

Reviewer #4: All comments have been addressed

2. Is the manuscript technically sound, and do the data support the conclusions?

Reviewer #3: Yes

Reviewer #4: Yes

3. Has the statistical analysis been performed appropriately and rigorously? 

Reviewer #3: Yes

Reviewer #4: Yes

4. Have the authors made all data underlying the findings in their manuscript fully available?

Reviewer #3: Yes

Reviewer #4: Yes

5. Is the manuscript presented in an intelligible fashion and written in standard English?

Reviewer #3: Yes

Reviewer #4: Yes

6. Review Comments to the Author

Reviewer #3: Congratulations on the authors for the improvement of the paper. They revised the paper according to comments of the reviewers.

Reviewer #4: Dear Authors,

Thank you very much to the authors for the thorough revisions and responses.

Kind regards,

7. PLOS authors have the option to publish the peer review history of their article (what does this mean?). If published, this will include your full peer review and any attached files.

Reviewer #3: **Yes: **Ferhat Esatbeyoglu

Reviewer #4: No

---

## [Editor Report · Acceptance letter]

10 Jan 2025

PONE-D-24-41589R1 

PLOS ONE

Dear Dr. Liu, 

I'm pleased to inform you that your manuscript has been deemed suitable for publication in PLOS ONE. Congratulations! Your manuscript is now being handed over to our production team.

Kind regards, 

on behalf of

Dr. Reza Rostamzadeh 

Academic Editor

PLOS ONE